# Prevalence and Risk Factors of Depression and Anxiety among Women in the Last Trimester of Pregnancy: A Cross-Sectional Study

**DOI:** 10.3390/medicina59061009

**Published:** 2023-05-24

**Authors:** Anca Ioana Cristea Răchită, Gabriela Elena Strete, Andreea Sălcudean, Dana Valentina Ghiga, Flavia Rădulescu, Mihai Călinescu, Andreea Georgiana Nan, Andreea Bianca Sasu, Laura Mihaela Suciu, Claudiu Mărginean

**Affiliations:** 1Doctoral School, “George Emil Palade” University of Medicine, Pharmacy, Science and Technology from Târgu Mureș, 540139 Târgu Mureș, Romania; anca.rachita@umfst.ro; 2Department of Psychiatry, “George Emil Palade” University of Medicine, Pharmacy, Science and Technology from Târgu Mureș, 540136 Târgu Mureș, Romania; 3Mental Health Center, Mureș County Clinical Hospital, 540072 Târgu Mureș, Romania; 4Department of Ethics and Social Sciences, “George Emil Palade” University of Medicine, Pharmacy, Science and Technology from Târgu Mureș, 540136 Târgu Mureș, Romania; 5Department of Medical Scientific Research Methodology, “George Emil Palade” University of Medicine, Pharmacy, Science and Technology from Târgu Mureș, 540136 Târgu Mureș, Romania; dana.ghiga@umfst.ro; 6Department of Endocrinology, “George Emil Palade” University of Medicine, Pharmacy, Science and Technology from Târgu Mureș, 540136 Târgu Mureș, Romania; flavia.maria1290@gmail.com; 7Graduate of Cluj School of Public Health, Babes-Bolyai University Cluj Napoca, 400347 Cluj-Napoca, Romania; mihai.calinescu.spring@gmail.com; 8First Department of Psychiatry, Clinical County Hospital, 540139 Târgu Mureș, Romania; nanandree96@yahoo.com (A.G.N.); andreea_vinteler@yahoo.com (A.B.S.); 9Department of Obstetrics and Gynecology Clinic II, “George Emil Palade” University of Medicine, Pharmacy, Science and Technology from Târgu Mureș, 540136 Târgu Mureș, Romania; laura.suciu@umfst.ro (L.M.S.); claudiu.marginean@umfst.ro (C.M.)

**Keywords:** prenatal, anxiety, depression, pregnancy, vulnerability, distress

## Abstract

Pregnancy represents a psychologically and emotionally vulnerable period, and research indicates that pregnant women have a higher prevalence of symptoms of anxiety and depression, debunking the myth that hormonal changes associated with pregnancy protect the mother. In recent years, several researchers have focused on the study of prenatal anxiety/depression—emotional disorders manifested by mood lability and low interest in activities—with a high prevalence. The main objective of this research was to conduct an antenatal screening in a cohort of pregnant women hospitalized for delivery in order to assess the prevalence of anxiety and depression. The secondary objective was to identify the risk factors associated with depression and anxiety in women in the third trimester of pregnancy. We carried out a prospective study in which we evaluated 215 pregnant women in the third trimester of pregnancy hospitalized for childbirth at the Obstetrics and Gynecology Clinic of the Târgu-Mureș County Clinical Hospital. The research was carried out between December 2019 and December 2021. The results showed that age and the environment of origin are the strongest predictors of mental health during pregnancy (OR = 0.904, 95%CI: 0.826–0.991; *p* = 0.029). For women from urban areas, there is an increased probability of falling at a higher level on the dependent variable (moderate depression) (OR = 2.454, 95%CI: 1.086–5.545; *p* = 0.032). In terms of health behaviors, none of the variables were statistically significant predictors of the outcome variable. The study highlights the importance of monitoring mental health during pregnancy and identifying relevant risk factors to provide appropriate care to pregnant women and the need for interventions to support the mental health of pregnant women. Especially in Romania, where there is no antenatal or postnatal screening for depression or other mental health conditions, these results could be used to encourage the implementation of such screening programs and appropriate interventions.

## 1. Introduction

Pregnancy represents a psychologically and emotionally vulnerable period, and research indicates that pregnant women have a higher prevalence of symptoms of anxiety and depression, debunking the myth that hormonal changes associated with pregnancy protect the mother. Pregnancy and the postpartum period are accepted by the scientific community as periods of increased circumstantial vulnerability to developing psychological distress [1].

Depression and anxiety are prevalent mental health conditions that can affect women during pregnancy, particularly in the last trimester. Several contributing factors have been identified that may increase the likelihood of experiencing these conditions during pregnancy. These can include a history of mental health disorders, personal or family history of depression or anxiety, previous traumatic life experiences, high levels of stress, lack of social support, and financial difficulties [2].

In recent years, there has been a growing interest among researchers in studying prenatal anxiety and depression, which are emotional disorders characterized by mood instability and decreased interest in activities. The prevalence of these disorders during pregnancy has been reported to range from 10 to 29.6% [3,4]. For perinatal depression, the global prevalence is estimated at 19.2% [5], with meta-analysis showing that the prevalence tends to be higher in the second and third trimesters (12.4%) than in the first trimester (7.4%) [6].

As such, prenatal anxiety and depression represent significant mental health concerns for pregnant women and their offspring. Identifying the risk factors, causes, and consequences of these disorders is critical for developing effective interventions to mitigate their negative impact.

Mothers from resource-limited environments may be particularly vulnerable to perinatal depression, with the prevalence of this condition potentially even higher in this group [7]. This could be attributed to various factors, such as low income and inadequate healthcare systems, which can act as risk factors for maternal psychopathology. Importantly, it should be noted that women who exhibit symptoms of depression and anxiety both before and during pregnancy may have an elevated risk of postnatal depression [8], which can adversely impact the child’s development. Therefore, it is crucial to identify and address these mental health conditions among pregnant women, especially in resource-limited settings where they may be more prevalent and have more significant consequences. Adequate healthcare access and appropriate interventions can help mitigate the risks of perinatal depression and its sequelae for both the mother and child [6]. According to a recent review, prenatal depression emerges as the most potent predictor of postnatal depression [9]. Non-psychotic depression is associated with sadness, hopelessness, sleep disturbances, fatigue, loss of appetite, feelings of worthlessness, lack of concentration, low self-esteem, and numerous neurotic-phobic symptoms [9].

Several studies have posited that the activation of the maternal stress response and alterations in the maternal endocrine and inflammatory systems have a significant role in the etiology of the effects on pregnancy and child development [10,11,12,13,14]. To date, a substantial body of literature has shown that stress during pregnancy can have an adverse impact on both pregnancy outcomes and the physiological development and behavior of offspring [10,15,16,17].

Notably, in most Eastern nations, prenatal check-ups primarily prioritize the physical health of pregnant women, with comparatively lesser emphasis placed on their emotional and mental health conditions. The prevalence of perinatal depression in Central and Eastern Europe is poorly known [18,19].

In Romania, there is no antenatal screening for depression or other mental health conditions, nor is there any screening in the postpartum period [20,21].

### Objectives

The main objective of this research was to conduct an antenatal screening in a cohort of pregnant women hospitalized for delivery, in order to assess the prevalence of anxiety and depression.

The secondary objective was to identify the risk factors associated with depression and anxiety in women in the third trimester of pregnancy.

It is important to understand the prevalence of perinatal depression and find a screening method that is quick and easy to administer so that it can be used in prenatal and postnatal family care. Early identification of perinatal depression is essential to ensure proper care for both mother and child, as this condition can have serious consequences if left untreated. However, perinatal depression is a treatable condition and there is evidence that treatment can be beneficial for both mother and child. Therefore, it is important to act accordingly to prevent and treat it.

## 2. Materials and Methods

### 2.1. Study Design

We carried out a prospective study in which we evaluated 215 pregnant women in the third trimester of pregnancy, hospitalized for childbirth, at the Obstetrics and Gynecology Clinic of the Târgu-Mureș County Clinical Hospital. The research was carried out between December 2019 and December 2021.

The research was conducted following the guidelines outlined in the Declaration of Helsinki and was approved by the Ethics Committee of the “George Emil Palade” University of Medicine, Pharmacy, Science, and Technology in Târgu Mureș. (No. 199/14/06/2019).

### 2.2. Participants and Procedure

A total of 215 patients expressed their desire to participate in the study, selected based on the following inclusion criteria: age ≥ or equal to 16 years, with a minimum level of education (be able to read and understand a text), third trimester of pregnancy (week 28–40), the patient’s consent.

The demographic questionnaire developed by us included 11 items, including demographic and health data (age of patients ≥ 16 years, the environment of origin, marital status, educational status, social statute, pregnancy type, gestational age, the number of pregnancies, complications arising in the third trimester of pregnancy, tobacco use, alcohol consumption).

Trained and qualified staff members distributed all questionnaires to participants, and each patient was provided with information about the importance and implications of the study, as well as their rights to data protection. Informed consent was obtained from all participants prior to their inclusion in the study. This ethical approach ensured that participants were fully informed and empowered to make an informed decision about their participation in the study and that their privacy and confidentiality were protected throughout the research process.

As exclusion criteria, we considered the following aspects: patients who did not fill in the full questionnaire, and patients with a history of mental disorders (patients with a past history of mental disorders may be considered as potential risk factors for the development of antenatal depression).

Because anxiety and depression are frequently associated disorders, almost impossible to distinguish from each other, they are assessed using scales. We opted for the following scales: HADS and EDPS for screening, and the Hamilton scale (anxiety/depression) for severity assessment.

### 2.3. Measures

Due to the fact that the patients were hospitalized for childbirth, we used the Hospital Anxiety and Depression Scale (HADS), developed by Zigmond and Snaith in 1983 [22], for screening depression and anxiety.

The Romanian version of the HADS scale presents adequate reliability and validity and was first created by Dr. Radu Teodorescu and published in *Sinapse* magazine (1996). In Romania, the validity of HADS has been confirmed both among adult psychiatric and medical patients, and validation studies have shown the high internal consistency of both subscales: HADS-A (Cronbach’s α ranges from 0.68 to 0.93) and HADS-D (Cronbach’s α ranges from 0.67 to 0.90). The scale was validated in the Romanian psychiatric population by Dr. Maria Ladea [23] in her doctoral thesis and includes 14 items regarding the intensity of anxiety and depression feelings. Each item is evaluated on a 4-point Likert scale, from 0 = never to 3 = always. A score between 0 and 7 indicates the absence of depression/anxiety, a score between 8 and 10 indicates mild symptoms, while a score above 11 indicates the presence of a case that requires additional attention. For patients who scored above 11, it was necessary to differentiate anxiety from depression, using the additional Hamilton Anxiety Scale (HAMA) and Hamilton Depression Rating Scale (HDRS). As reference values, we consider that a score below 6 indicates a non-clinical case, while a score above 6 indicates a clinical case.

For the assessment of depression severity, we utilized the Hamilton Depression Rating Scale (HAMD), also known as the Hamilton Depression Rating Scale [24]. This scale quantifies the severity of depression in patients who have already received a diagnosis of depression. In clinical practice, the 17-item version is commonly used as Hamilton himself recommended that the last four items (diurnal variation, depersonalization/derealization, paranoid symptoms, and obsessive-compulsive symptoms) not be included in the total score, as these symptoms are not reflective of the overall severity of depression [25] Hamilton constructed the scale to quantify information from the patient, taken by an experienced clinician based on an unstructured interview (without specifying questions or evidence). The total score represents the following: 0–7 absence, 8–16 mild, 17–22 moderate, ≥23 severe.

Although some items on the scale are rated on a scale of 0 to 8, while others are rated on a scale of 0 to 4, the total score of the 17 items has a strong correlation of 0.93 with the first-factor score. This demonstrates that the scale is effective in providing a straightforward method for measuring the severity of a patient’s condition and tracking changes in their condition over time [26].

The Hamilton Anxiety Scale (HAMA) [27] was employed to gauge the severity of anxiety. The scale consists of a semi-structured interview with 14 items that assess both cognitive and somatic symptoms of anxiety, and the maximum score is 56. The examiner assigns scores from 0 to 4 to each item. The scores represent the following: 0—absent, indicating that the subject has never experienced these symptoms; 1—weak, indicating that symptoms occur irregularly and for brief periods; 2—moderate, indicating that symptoms manifest consistently for certain periods, requiring the subject’s effort to deal with them; 3—severe, indicating that symptoms are continuous and dominate the subject’s life; and 4—very severe, indicating that symptoms are disabling and impede the subject’s activities. The total score represents the following: 0–4 normal, 5–10 mild, 11–16 moderate, ≥17 severe. The scale permits a comprehensive evaluation of both psychological (e.g., mental stress, anxiety state) and somatic (e.g., associated biophysiological changes) symptoms of anxiety. Through factor analysis, items were determined that correspond to the mental and somatic aspects of anxiety. Expert studies have revealed that individuals diagnosed with generalized anxiety disorder or panic attacks tend to receive high scores (above 20) on the HAMA, whereas those without a clinical diagnosis tend to receive significantly lower scores [28].

The EPDS was originally developed to help identify possible depressive symptoms in the postnatal period. In addition to the HAM-D17, we employed the Edinburgh Postnatal Depression Scale (EDPS), a self-report questionnaire consisting of 10 items and originally designed for research purposes, to facilitate the identification of perinatal depression. This tool has been available for over three decades and is widely used for detecting symptoms of depression during and after pregnancy [29].

The study utilized the Edinburgh Postnatal Depression Scale (EPDS) as a self-report questionnaire to detect depressive symptoms during the antenatal period with adequate sensitivity and specificity. Responses on the EPDS are scored on a 4-point scale (0–3), with higher scores indicating a greater endorsement of each symptom. Participants with a total score of 12 or more on the EPDS were classified as having a high possibility of depression, while a score of 9–11 indicated a possible risk of depression. The study employed the Romanian version of the EPDS (EPDS-R) [30], which has been validated for use in the Romanian population. The mean EPDS-R score was 10.1 (SD: 6.5), with scores ranging from 0 to 29.

The study also utilized the Center for Epidemiologic Studies Depression Scale (CES-D) as a self-report questionnaire to assess depression. The Romanian version of the CES-D (CES-D(R)) was used. The mean EPDS-R score was 10.1 (SD: 6.5), with scores ranging from 0 to 29. The mean CESD(R) score was 24.7 (SD: 13.1), with scores ranging from 7 to 60. As indicated by Cronbach’s α, reliability (internal consistency) for both EPDS-R and CES-D(R) scores were robust (α = 0.88 and 0.93, respectively) in this study sample.

The Romanian version of the Edinburgh Postnatal Depression Scale (EPDS-R) has been validated for use in detecting depressive symptoms in pregnant women in Romania. In this version, a score of >12 was identified as the optimal reference point for prenatal screening, with a correct classification rate of 81.1%. Using this cutoff point, the prevalence rate of depressive symptoms in the study sample was found to be 33.7%.

The EPDS-R demonstrated robust psychometric properties in the population and is recommended as a valid, reliable, and easy-to-administer tool for detecting depression in pregnant women in Romania. The reference values for the EPDS-R in this study were nonclinical (<10), subclinical (10–11), and clinical (≥12), as defined by the original EPDS [31].

Symptom severity was based on cutoff scores and ranked (EPDS, 1 = no depression, 2 = possible depression, 3 = high possibility of depression, and 4 = depressed; GAD, 1 = minimal anxiety, 2 = mild anxiety, 3 = moderate anxiety and 4 = severe anxiety).

### 2.4. Statistical Analysis

The statistical analysis included both descriptive (frequency, percentage) and inferential statistics. We applied the Chi-square test and the Fisher test to determine the association of qualitative variables. To quantify the relationship between multiple predictor variables and the outcome variable, we performed binary logistic regression and ordinal regression for ordinal dependent variables. For each independent variable included, the estimated coefficient (estimate) associated with each independent variable, the 95% confidence interval around the respective estimate, the exponential coefficient (Exp (B)), the Wald 95% confidence interval around the exponential coefficient, and the *p*-value associated with that independent variable were calculated. The *p*-value represents the statistical significance of the relationship between each independent variable and the dependent variable. The significance threshold was chosen for a *p*-value of 0.05. Microsoft Excel 2016 (Microsoft^®^ Corp., Redmond, WA, USA) was used for data entry and analyzes were analyzed using IBM SPSS Statistics [30,32].

## 3. Results

A total of 215 pregnant women in the third trimester were included in the study after applying the exclusion criteria. The age range of participants was 16 to 40 (mean = 26.35). The majority (55.35%) were between 25 and 34 years old, with an average of the groups of women ≤ 35 years old of between 21.22472–28.14286, while in the ≥35-years-old group, the average was 37.42857. The majority of subjects (53.49%) were from urban areas. Regarding marital status, the majority (85.12%) were single/divorced, and 91.63% had no higher education. Regarding social status, the majority (67.91%) were employed. In terms of pregnancy planning, 87.44% were unplanned, while in terms of the number of previous pregnancies, 62.79% were multiparous. In terms of nicotine and alcohol consumption, almost half of the women (49.77%) reported smoking, while one-quarter (22.79%) reported alcohol consumption (Table 1).

The characteristics in Table 1 represent the frequency and percentage of pregnant women included in the study according to socio-demographic variables and information related to their health.

The presence of depression, anxiety is demonstrated in Table 2, which shows that 1/3 (31.63%) of the participants scored in the clinical range for depression on the EDPS scale, while 55.35% scored in the nonclinical range and 13.02% scored in the subclinical range.

Anxiety measured on the HADS A scale highlights a majority (88.37%) with scores in the clinical range and a small proportion (11.63%) in the non-clinical range.

Regarding depression measured on the HADS D scale, approximately the same majority (84.65%) scored in the clinical range, and a small proportion (15.35%) in the nonclinical range.

We established associations between depression scores (EDPS, HADS A, and HADS D) and various socio-demographic and health factors. To take into account the complexity and to have more ordered levels in the dependent variables, we used ordinal regression, allowing the interaction of these variables with one or more independent variables.

Table 3, Table 4, Table 5, Table 6 and Table 7 show the results of the logistic regression analysis with the variables EDPS (a measure of depression or psychological distress), HADS A, HADS D, and several predictor variables including age, background, marital status, level of education, employment status, pregnancy characteristics, and health behaviors (smoking, alcohol consumption). For each independent variable, the estimated coefficient (estimate) associated with each independent variable, the 95% confidence interval around the respective estimate, the exponential coefficient (Exp (B)), the Wald 95% confidence interval around the exponential coefficient, and the *p*-value associated with that independent variable were included. The *p*-value represents the statistical significance of the relationship between each independent variable and the dependent variable.

Based on the *p*-values, for EDPS (Table 3), none of the variables is a statistically significant predictor of the outcome variable.

Based on the *p*-values, for HAMD (Table 4), age and background are statistically significant predictors of the outcome variable. As age increases there is a decreased probability of falling at a higher level on the dependent variable (moderate depression) (OR = 0.904, 95%CI: 0.826–0.991; *p* = 0.029). For women from urban areas, there is an increased probability of falling at a higher level on the dependent variable (moderate depression) (OR = 2.454, 95%CI: 1.086–5.545; *p* = 0.032).

Based on the *p*-values, for HAMA (Table 5), the environment of origin is a statistically significant predictor of the outcome variable. For women from urban areas, there is an increased probability of falling to a higher level on the dependent variable (moderate depression), (OR = 1.845, 95%CI: 1.004–3.391), statistically significant (*p* = 0.046).

Based on the *p* values, for HADS D and HADS A (Table 6 and Table 7), none of the variables are statistically significant predictors of the outcome variable.

## 4. Discussion

We tried to assess how widespread antenatal anxiety and depression are in a sample of pregnant women in the last trimester of pregnancy using the following scales: EDPS, HADS A, HADS D, HAMA, and HAMD. The study was carried out on a small sample of pregnant women in the third trimester of pregnancy hospitalized for childbirth from a geographical area limited to a single Mures county, (with the particularity: according to the data of the National Institute of Statistics (INNS), in Romania, in 2019 the number of divorces increased by 1.7 compared to the previous year, the number of registered marriages was decreasing, and last but not least, there was an increase in the number of teenage mothers (16–20 years old), which may limit the generalization of the results to the Romanian population [33].

The study found a prevalence of prenatal depression of 31.63% (Table 2), which is similar to what has been reported in other studies.

Studies conducted in Turkey, Canada, India, and Nepal have also reported a significant prevalence of depressive symptoms in pregnant women [34,35,36,37]. For example, a Canadian study [35] with 1987 participants reported a prevalence of depressive symptoms in 37% of women and anxiety symptoms in 56.6%. Similarly, Sanchana et al. [35] reported a prevalence of 40.5% and Sapkota et al. [36] found that 37.5% of participants had some level of depressive symptoms. In a recent study by Umuziga [37], it was found that 37.6% of women in their third trimester of pregnancy had suggestive depression symptoms (EPDS ≥ 10). These findings suggest that prenatal depression is a significant health issue affecting a substantial proportion of pregnant women worldwide.

In this study, the prevalence of anxiety symptoms in pregnant women was found to be high, with 88.37% (Table 2) of participants presenting scores in the clinical range on the HADS A scale. A score below 6 indicating a nonclinical case was used as a reference, with a score above 6 indicating a clinical case; therefore, the values obtained were high (Table 2). This finding is consistent with previous studies in the literature that have reported high prevalence rates of anxiety in pregnant women worldwide. For example, Leach et al. (2014) [38] found that prevalence rates of anxiety symptoms in pregnant women in Australia ranged from 6.8% to 59.5%. Similarly, studies in Canada (Dennis et al., 2017; Fawcett et al., 2019) [38,39] and the United Kingdom (Nielsen-Scott et al., 2019) [40] have also reported high prevalence rates of anxiety in pregnant women. It should be noted that differences in prevalence rates may be due to various factors, including the assessment tools and cutoff points we used, as well as cultural differences in the perception of mental health.

We also examined the influence of socio-demographic variables and mental health status (Table 1).

The study found that depressive symptoms during pregnancy, as measured by the HAMD scale, are associated with factors such as age and environment. Additionally, the results showed that younger maternal age is statistically associated with higher levels of prenatal depressive symptoms on the HAMD scale. However, there is inconsistency in the literature regarding this association, as some studies have found that younger maternal age increases the likelihood of depression during pregnancy. This is not surprising given that other similar studies have also shown that younger women are more likely to develop prenatal depression [41,42,43].

The association between age and antenatal depression is a topic of interest for researchers, and findings have been inconsistent. We found that the age range of participants was 16 to 40 (mean = 26.35), with the majority (55.35%) being between 25 and 34 years old. Some studies suggest that younger age is associated with a higher risk of depression during pregnancy, particularly for women under 25 years old [44], under 20 years old [45], or between 15 and 20 years old [46]. This may be due to factors such as incomplete education, unstable economic situations with lower income and job instability, and unemployment [46]. Conversely, other studies have found that older maternal age is associated with a higher risk of antenatal depression, particularly for women over 35 years old or over 30 years old [47,48].

Research suggests that environment may be a significant factor in the development of prenatal depressive symptoms, but findings are inconclusive in many studies. Regarding our study, we confirmed that women from urban areas have a statistically increased risk of developing prenatal depressive symptoms and anxiety. In contrast, other studies suggest that women in rural areas are at increased risk of risk factors for antenatal depression, such as obesity and diabetes [49,50,51,52].

Nidey et al. [53], mention that women from a rural environment have an increased risk of perinatal depression compared to women from an urban environment. This suggests that women from a rural environment may experience unique health barriers that may increase the risk of depression. The results highlight the importance of identifying and prioritizing women at risk of perinatal depression and anxiety and ensuring access to appropriate interventions and support.

Civil status was not found to be a risk factor for depression and anxiety, despite 85.12% being divorced/unmarried (Table 1). This can be explained by the evolution of Romanian society and the changing role of women in it: the change in values and social perceptions about family and marriage, the increase in women’s financial independence, more young women choosing to have children before getting married compared to previous generations, a desire to avoid potential relationship problems, and some may consider marriage as an outdated social convention and not a priority in their lives.

Other studies have shown that partner absence, including being single, divorced, separated, or in polygamous marriages, is associated with higher rates of depression during pregnancy [54].

In contrast, our results did not find a significant association between a low educational level and depression and anxiety during pregnancy, but previous research has consistently shown a link between lower educational levels and antenatal depression [55,56,57,58,59,60].

In our research, we also analyzed the social status factor in relation to depression and anxiety during pregnancy, but we did not identify any association between them. However, various studies have shown that poor socioeconomic status is often associated with the occurrence of prenatal depression [61,62,63,64]. This is often due to the economic difficulties encountered in such situations, which can form a vicious circle for antenatal depression.

The mental well-being of pregnant women is of utmost importance, which is why researchers need to consider obstetric factors when studying the risk of antenatal depression [65,66,67]. Unplanned pregnancy has been found in several studies to be a risk factor for antenatal depression, while planned pregnancy has been associated with lower levels of depression. In terms of the type of pregnancy, our study did not find significant associations with depression and anxiety, but other researchers have consistently shown that women who did not plan their pregnancy are more prone to depression [49,68]. The relationship between the number of pregnancies and antenatal depression is still unclear, with conflicting results in different studies. While some studies found that nulliparity is associated with an increased risk of depression during pregnancy [47], most studies suggest that multiparity is linked to higher levels of antenatal depression [47,67,68,69].

Another obstetric factor studied in relation to depression and anxiety was gestational age. In our study, gestational age was not a predictive factor for depression or anxiety. However, studies such as the one by Rezaee R. and Framarzi M. demonstrated a positive correlation between gestational age and anxiety (F = 1.903, *p* = 0.006) and depression (F = 2.101, *p* = 0.003) in a proportion of 68.1% [70].

Smoking and alcohol consumption during pregnancy may increase the risk of depression and the relationship may be mediated by oxidative stress. In terms of nicotine and alcohol consumption, almost half of the women (49.77%) reported smoking, while one-quarter (22.79%) reported alcohol consumption (Table 1). From the point of view of health behaviors (smoking, alcohol consumption) we calculated the estimated coefficient associated with each independent variable, but none of the variables was a statistically significant predictor of the outcome variable (Table 3). Other authors [71] identified interaction effects between tobacco use and childbirth anxiety (*p* < 0.001), and interaction between tobacco use and pregnancy depression (*p* = 0.032), demonstrating that there are different types of interaction effects between tobacco use and anxiety or depression. In addition, the study of Wubetu et al. [72] suggests that alcohol consumption during pregnancy may increase the risk of depression and that this relationship may be mediated by a number of factors, such as social support, anxiety, and stress. Considering the negative impact of alcohol and tobacco consumption during pregnancy, more importance should be given to primary prevention. It is very important to carry out periodic assessments throughout pregnancy, as they can help identify health problems or early intervention in situations that require medical attention.

### Limits of the Study

This study conducted on pregnant women in the third trimester was limited by: a small sample size, the inclusion of only women hospitalized for childbirth, and a limited geographic area, which restricts the generalizability of the findings to the general population. Furthermore, the study did not consider the impact of stressful life events, such as the COVID-19 pandemic, and other psychosocial risk factors, such as social support, couple relationships, a family history of mental disorders, and personality traits. These factors could have affected the accuracy and comprehensiveness of the study’s results.

It is important to acknowledge these limitations when interpreting the study’s findings and to exercise caution when extrapolating them to other populations or contexts. Future research with larger and more diverse samples, and a broader range of factors considered, could provide a more complete understanding of the relationship between pregnancy and mental health.

## 5. Conclusions

The study underscores the significance of monitoring mental health throughout pregnancy, as well as identifying pertinent risk factors in order to deliver appropriate care to expectant mothers.

The manifestation of both anxiety and depression during pregnancy is often linked to several factors. Recognizing these associated factors can facilitate the early implementation of mental health monitoring measures throughout pregnancy.

Demographic and health-related factors are closely interrelated and can have a significant impact on the maternal psycho-affective state. Upon examining the associated risk factors, it was revealed that younger maternal age is associated with prenatal depressive symptoms. Anxiety was also found to be a common occurrence during pregnancy.

The results obtained showed that age and the environment of origin are the strongest predictors of mental health during pregnancy.

The introduction of a screening program that also provides psychological counseling in prenatal care can be very helpful for women who are experiencing stress and anxiety or who are in disadvantaged social situations. This program could be implemented at the level of the health system, and medical staff could be trained in providing appropriate counseling and support to these women.

Moreover, the findings emphasize the necessity for implementing interventions aimed at supporting the mental well-being of pregnant women.

## Figures and Tables

**Table 1 medicina-59-01009-t001:** Frequency according to socio-demographic variables and health-related information (N = 215).

Characteristics		Frequency (n)	%
**Socio-demographic**			
Age category	<25	89	41.40%
	25–34	119	55.35%
	≥35	7	3.26%
Environment	Rural	100	46.51%
	Urban	115	53.49%
Marital status	Married	32	14.88%
	Unmarried/Divorced	183	85.12%
Educational status	No higher education	197	91.63%
	With higher education	18	8.37%
Social status	Employed	146	67.91%
	Housewife	69	32.09%
**Health information**			
Pregnancy type	Unplanned	188	87.44%
	Planned	27	12.56%
Gestational age	29–33	28	13.02%
	≥34	187	86.98%
Number of pregnancies	Multiparous	135	62.79%
	Nulliparous	80	37.21%
Smoking	Nonsmoking	108	50.23%
	Smoking	107	49.77%
Alcohol consumption	No alcohol consumption	166	77.21%
	With alcohol consumption	49	22.79%
	**Total**	**215**	**100.00%**

Legend: n = number; %—percentage.

**Table 2 medicina-59-01009-t002:** Proportion of participants with nonclinical, clinical, and subclinical depression/anxiety (Chi-square test).

**EDPS**	**Frequency (n)**	**%**
Clinical	68	31.63%
Nonclinical	119	55.35%
Subclinical	28	13.02%
Total	215	100.00%
**HADS A**	**Frequency (n)**	**%**
Clinical	190	88.37%
Nonclinical	25	11.63%
Total	215	100.00%
**HADS D**	**Frequency (n)**	**%**
Clinical	182	84.65%
Nonclinical	33	15.35%
Total	215	100.00%

Legend: n = number; %—percentage; EDPS (Edinburgh Postnatal Depression Scale): Nonclinical < 10, Subclinical, 10–11, Clinical ≥ 12; HADS A (Hospital Anxiety and Depression Scale): No anxiety under 6 (nonclinical), anxiety over 6 (clinical); HADS D (Hospital Anxiety and Depression Scale) No depression under 6 (nonclinical), depression over 6 (clinical).

**Table 3 medicina-59-01009-t003:** Association of depressive symptoms (EDPS) with socio-demographic and health factors (ordinal logistic regression) [32].

EDPS		Estimate	95% C.I.		Exp (B)	95% Wald C.I. for Exp (B)		Value *p*
			Lower	Upper		Lower	Upper	
Predictors	Age	−0.021	−0.092	0.05	0.98	0.911	1.053	0.569
	Environment	0.314	−0.316	0.944	1.369	0.726	2.580	0.329
	Marital status	−0.295	−1.073	0.482	0.744	0.339	1.636	0.457
	Educational Status	−0.356	−1.335	0.623	0.701	0.262	1.871	0.476
	Social Status	−0.277	−0.929	0.375	0.758	0.391	1.469	0.405
	Pregnancy type	0.049	−0.817	0.914	1.050	0.44	2.504	0.912
	Pregnancy age	0.268	−0.573	1.108	1.307	0.559	3.059	0.532
	No. of pregnancies	−0.084	−0.715	0.548	0.92	0.484	1.747	0.795
	Smoker	−0.386	−0.925	0.153	0.68	0.397	1.163	0.161
	Alcohol consumption	0.122	−0.539	0.783	1.130	0.584	2.186	0.717

Legend: 95% C.I.—confidence interval; Exp (B)—exponential coefficient; EDPS (Edinburgh Postnatal Depression Scale): Nonclinical < 10, Subclinical, 10–11, Clinical ≥ 12; Obs. No predicted change in the probability of being in a higher category as the values of the independent variables increase.

**Table 4 medicina-59-01009-t004:** Association of depressive symptoms (HAMD) with socio-demographic and health factors (ordinal logistic regression) [32].

HAMD		Estimate	95% C.I.		Exp (B)	95% Wald C.I. for Exp (B)		Value *p*
			Lower	Upper		Lower	Upper	
Predictors	Age	−0.101	−0.191	−0.01	0.904	0.826	0.991	0.029
	Environment	0.898	0.078	1.717	2.454	1.086	5.545	0.032
	Marital status	−0.641	−1.678	0.396	0.527	0.184	1.505	0.225
	Educational Status	−0.948	−2.234	0.338	0.388	0.109	1.383	0.149
	Social Status	−0.039	−0.862	0.785	0.962	0.42	2.202	0.927
	Pregnancy type	0.284	−0.817	1.386	1.329	0.449	3.931	0.613
	Pregnancy age	0.776	−0.238	1.790	2.173	0.789	5.988	0.134
	No. of pregnancies	0.783	−0.026	1.592	2.188	0.972	4.925	0.058
	Smoker	−0.327	−1.019	0.365	0.721	0.361	1.438	0.354
	Alcohol consumption	0.329	−0.536	1.194	1.390	0.59	3.274	0.456

Legend: 95% C.I.—confidence interval; Exp (B)—exponential coefficient; HAMD (Hamilton Depression Rating Scale): 0–7 absence, 8–16 mild, 17–22 moderate, ≥23 severe.

**Table 5 medicina-59-01009-t005:** Association of depressive symptoms (HAMA) with socio-demographic and health factors (ordinal logistic regression) [32].

HAMA		Estimate	95% C.I.		Exp (B)	95% Wald C.I. for Exp (B)		Value *p*
			Lower	Upper		Lower	Upper	
Predictors	Age	0.009	−0.06	0.077	1.009	0.944	1.078	0.803
	Environment	0.613	0.01	1.215	1.845	1.004	3.391	0.046
	Marital status	0.236	−0.513	0.986	1.267	0.615	2.610	0.537
	Educational Status	0.239	−0.721	1.198	1.270	0.481	3.354	0.626
	Social Status	−0.158	−0.784	0.467	0.854	0.454	1.604	0.620
	Pregnancy type	0.028	−0.801	0.858	1.029	0.451	2.347	0.947
	Pregnancy age	0.134	−0.648	0.916	1.143	0.529	2.471	0.737
	No. of pregnancies	0.424	−0.186	1.033	1.527	0.829	2.813	0.173
	Smoker	−0.365	−0.884	0.155	0.695	0.413	1.169	0.169
	Alcohol consumption	−0.236	−0.871	0.4	0.79	0.414	1.507	0.468

Legend: 95% C.I.—confidence interval; Exp (B)—exponential coefficient; HAMA (Hamilton Anxiety Rating Scale): 0–4 normal, 5–10 mild, 11–16 moderate, ≥17 severe.

**Table 6 medicina-59-01009-t006:** Association of depressive symptoms (HADS D) with socio-demographic and health factors (ordinal logistic regression) [32].

HADS-D		Exp (B)	95% C.I. for Exp (B)		Value *p*
			Lower	Upper	
Predictors	Age	1.028	0.922	1.145	0.620
	Environment	0.582	0.233	1.450	0.245
	Marital status	1.013	0.302	3.400	0.983
	Educational Status	1.213	0.305	4.832	0.784
	Social Status	1.351	0.532	3.432	0.527
	Pregnancy type	2.148	0.718	6.428	0.172
	Pregnancy age	1.654	0.592	4.623	0.337
	No. of pregnancies	1.168	0.473	2.885	0.737
	Smoker	0.907	0.415	1.982	0.807
	Alcohol consumption	1.583	0.583	4.297	0.367

Legend: Exp (B)—exponential coefficient; 95% C.I.—confidence interval; HADS-D (Hospital Anxiety and Depression Scale): Nonclinical under 6, Clinical over 6.

**Table 7 medicina-59-01009-t007:** Association of depressive symptoms (HADS A) with socio-demographic and health factors (Ordinal logistic regression) [32].

HADSA		Exp (B)	95% C.I. for Exp (B)		Value *p*
			Lower	Upper	
Predictors	Age	1.035	0.914	1.171	0.590
	Environment	2.409	0.85	6.825	0.098
	Marital status	1.091	0.314	3.790	0.891
	Educational Status	1.698	0.316	9.126	0.537
	Social Status	1.149	0.416	3.173	0.789
	Pregnancy type	0.743	0.196	2.813	0.661
	Pregnancy age	1.134	0.323	3.979	0.844
	No. of pregnancies	1.157	0.421	3.174	0.778
	Smoker	0.696	0.287	1.689	0.423
	Alcohol consumption	0.472	0.179	1.241	0.128

Legend: Exp (B)—exponential coefficient; 95% C.I.—confidence interval; HADS A (Hospital Anxiety and Depression Scale): Nonclinical under 6, Clinical over 6.

## Data Availability

Data is unavailable due to privacy and ethical restrictions.

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
