# Peer review of "Prevalence and Risk Factors of Depression and Anxiety among Women in the Last Trimester of Pregnancy: A Cross-Sectional Study"

_medicina, 2023, doi:10.3390/medicina59061009_

Round 1

Reviewer 1 Report

Dear Editor/authors

This is an interesting survey about predictive factors and the prevalence of depression in pregnancy. Although the article is scientific and well-written, some changes can help improve it.

Title:

According to the main objective, which is to investigate depression and anxiety, it is better to use the term "mood" in the title or to add the term "anxiety" as well.

Abstract: 

Before reminding the purpose of the study, it is recommended to introduce the background of the study (anxiety and depression in pregnancy) in one paragraph.

Keywords:

It is recommended to select keywords from the MeSH to make it easier to retrieve the article during the search (for example prenatal, anxiety, depression)

Introduction:

Although important points are mentioned in the introduction, the content and paragraphs are fragmented. First, an explanation (introduction) should be given about the subject of the study, then the global and European situation, and after that, the situation in Romania should be explained. In the last step, the importance of the evaluating main subject and finally the purpose of the study should be stated.

I think it is necessary to change the place of some paragraphs. 

Materials and Methods:

1. In the exclusion criteria, were high-risk pregnancies, which are usually associated with stress, identified?

ï¼’. In the statistical analysis section, statistical tests and p-value are stated twice, which are better to be combined.

Results:

1. Less than half of the studied people (about 46%) are from rural areas. Why is it stated that more than 2/3?

More than 2/3 (46.51%) of them come from rural areas”

ï¼’. An important question that should be answered based on the results of marriage and education status is whether the sampling (methodology) was representative of the target population or the Romanian society.  

“Regarding marital status, the majority (85.12%) were single/divorced and 91.63% had no higher education”

Conclusion:

Some of the statements made are not related to the conclusions of this study and should be transferred to the suggestions section, such as the use of alcohol and tobacco and the need for periodic evaluations of pregnant women.

Author Response

Dear Reviewer,

   Thank you very much for your observations. Please find below the modifications made in the manuscript.

Title:

According to the main objective, which is to investigate depression and anxiety, it is better to use the term "mood" in the title or to add the term "anxiety" as well.

In accordance with your suggestion, we modified the title: Contributing factors and prevalence of depression as well anxiety among women in the last trimester of pregnancy.

Abstract: 

Before reminding the purpose of the study, it is recommended to introduce the background of the study (anxiety and depression in pregnancy) in one paragraph.

We introduced another paragraph on the background of the study: Pregnancy represents a psychologically and emotionally vulnerable period, and research indicates that pregnant women have a higher prevalence of symptoms of anxiety and depression, debunking the myth that hormonal changes associated with pregnancy protect the mother. In recent years, several researchers have focused on the study of prenatal anxiety/depression, emotional disorders manifested by mood lability, low interest in activities, with a prevalence ranging from 10 to 29.6%.

Keywords:

It is recommended to select keywords from the MeSH to make it easier to retrieve the article during the search (for example prenatal, anxiety, depression)

Thank you very much for your observation, we took the keywords from MeSH: prenatal, anxiety, depression, pregnancy, vulnerability, distress.

Introduction:

Although important points are mentioned in the introduction, the content and paragraphs are fragmented. First, an explanation (introduction) should be given about the subject of the study, then the global and European situation, and after that, the situation in Romania should be explained. In the last step, the importance of the evaluating main subject and finally the purpose of the study should be stated.

I think it is necessary to change the place of some paragraphs. 

Thank you very much for your observation, we made the necessary modification in the Introduction part of the manuscript.

Materials and Methods:

  • In the exclusion criteria, were high-risk pregnancies, which are usually associated with stress, identified?

As exclusion criteria we considered the following aspects: patients who did not fill in the full questionnaire, and patients with a history of mental disorders (patients with a past history of mental disorders may be considered as potential risk factors for the development of antenatal depression).

  • In the statistical analysis section, statistical tests and p-value are stated twice, which are better to be combined.

Thank you very much for your observation, we made the changes accordingly in the manuscript.

Results:

1. Less than half of the studied people (about 46%) are from rural areas. Why is it stated that more than 2/3?

More than 2/3 (46.51%) of them come from rural areas”

Thank you for your observation, we changed in the manuscript.

ï¼’. An important question that should be answered based on the results of marriage and education status is whether the sampling (methodology) was representative of the target population or the Romanian society.  

“Regarding marital status, the majority (85.12%) were single/divorced and 91.63% had no higher education”

Thank for your observation, we answered accordingly in the manuscript.

Conclusion:

Some of the statements made are not related to the conclusions of this study and should be transferred to the suggestions section, such as the use of alcohol and tobacco and the need for periodic evaluations of pregnant women.

We made the necessary changes accordingly :  It is pertinent to highlight that the prevalence of depression and anxiety in pregnant women is known to vary across different cultures and countries. Furthermore, prevalence estimates may also differ depending on the assessment tools utilized. The manifestation of both anxiety and depression during pregnancy is often linked with several factors. Recognizing these associated factors can facilitate early implementation of mental health monitoring measures throughout pregnancy.

Demographic and health-related factors are closely interrelated and can have a sig-nificant impact on the maternal psycho-affective state. Upon examining the associated risk factors, it was revealed that younger maternal age is associated with prenatal depres-sive symptoms. Additionally, the prevalence of prenatal depression was similar to previ-ously reported rates in the literature. Anxiety was also found to be a common occurrence during pregnancy.

The results obtained showed that age and the environment of origin are the strongest predictors of mental health during pregnancy.

The introduction of a screening program that also provides psychological counseling in prenatal care can be very helpful, for women who are experiencing stress and anxiety or who are in disadvantaged social situations. This program could be implemented at the level of the health system, and medical staff could be trained in providing appropriate counseling and support to these women.

The study underscores the significance of monitoring mental health throughout pregnancy, as well as identifying pertinent risk factors in order to deliver appropriate care to expectant mothers. Moreover, the findings emphasize the necessity for implementing interventions aimed at supporting the mental well-being of pregnant women.

    Thank you very much for you time in reviewing our manuscript!

Reviewer 2 Report

The abstract needs to have a brief about the methods and the results (numbers)

Introduction

-         Needs English proof reading

-         For example line 75 : “objective”!!

-         The last two paragraphs in the introduction should be revised in their order

Methods

-         The study recruited 215 pregnant women, How was this sample size arrived at? How many were initially approached? Please elaborate on the exclusion criteria, i.e. women with mental health, what disturbances and why?

-         Line 104 “To investigate the presence or absence of depression/anxiety” this is inaccurate, these scales assess or evaluate or screen , please revise the language used

-         The write up of the study instrument needs to be rewritten, it should start with the demographics, clinical details followed by the scales

-         Also, the authors used 3 scales to screen depression ! why ? any rationale?

Results

-         Very un neat and missy write up

-         Very hard to read

-         The response rate should be present

-         Where is the t-test?

-         The ORs should be found in the text

-         The tables are hard to be understood, use a clear format, neat font and three columns OR (crude or adjusted), 95CI and p-value

-         Under each table, add an appropriate legend

Discussion

The discussion is not aligned with the objectives

Needs massive work

The manuscript needs entire scientific and linguistic modifications

Author Response

Dear Reviewer,

   Thank you very much for your observations.  Please find below the answers to your observations.

Introduction

-         For example line 75 : “objective”!!

      Thank you very much for your observation, we made the necessary changes in the manuscript.

  -         The last two paragraphs in the introduction should be revised in their order

       Thank you very much for your observation, we changed the paragraphs in the Introduction part.

Methods

-         The study recruited 215 pregnant women, How was this sample size arrived at? How many were initially approached? Please elaborate on the exclusion criteria, i.e. women with mental health, what disturbances and why?

215 patients expressed their desire to participate in the study, who were selected based on the following inclusion criteria: age ≥ or equal to 16 years, with a minimum level of education (be able to read and understand a text), third trimester of pregnancy (week 28-40), the patient's consent.

The demographic questionnaire developed by us included 11 items, including demographic and health data (age of patients ≥ 16 years, the environment of origin, marital status, educational status, social statute, pregnancy type, gestational age, the number of pregnancies, complications arising in the third trimester of pregnancy, tobacco use, alcohol consumption).

Trained and qualified staff members distributed all questionnaires to participants, and each patient was provided with information about the importance and implications of the study, as well as their rights to data protection. Informed consent was obtained from all participants prior to their inclusion in the study. This ethical approach ensured that participants were fully informed and empowered to make an informed decision about their participation in the study, and that their privacy and confidentiality were protected throughout the research process.

And as exclusion criteria we considered the following aspects: patients who did not fill in the full questionnaire, and patients with a history of mental disorders (patients with a past history of mental disorders may be considered as potential risk factors for the development of antenatal depression). Informed consent was obtained from all participants before enrollment in the study.

Because anxiety and depression are frequently associated disorders, almost impossible to distinguish from each other, they are assessed using scales. We opted for the following scales: HADS and EDPS for screening, and the Hamilton scale (anxiety/depression) for severity assessment.

-         Line 104 “To investigate the presence or absence of depression/anxiety” this is inaccurate, these scales assess or evaluate or screen, please revise the language used.

    Thank you very much for your observation. We made the necessary changes in the manuscript.

-         The write up of the study instrument needs to be rewritten, it should start with the demographics, clinical details followed by the scales.

   We appreciate your observation and made the necessary changes in the manuscript.

-         Also, the authors used 3 scales to screen depression ! why ? any rationale?

We found it appropriate to use the 3 scales as anxiety and depression are frequently associated disorders. As it is almost impossible to distinguish from each other, they are assessed using scales. We opted for the following scales: HADS and EDPS for screening, and the Hamilton scale (anxiety/depression) for severity assessment.

Results

-         The response rate should be present

       Thank you very much for your observation. We made the necessary changes in the manuscript.

-         Where is the t-test?

We applied the Chi-square test and the Fisher test to determine the association of qualitative variables.

-         The ORs should be found in the text

        We introduced the requested information in the manuscript.

-         The tables are hard to be understood, use a clear format, neat font and three columns OR (crude or adjusted), 95CI and p-value

              We made the necessary modifications in the manuscript.

-         Under each table, add an appropriate legend

       We introduced the legend for teach table.

Discussion

The discussion is not aligned with the objectives

 Thank you very much for your observation. We modified the Discussion part.

    Thank you very much for your time in reviewing our manuscript!

Reviewer 3 Report

Thank you for this valuable contribution to knowledge about women and depression. Please respond to the following concerns:

2.3 Methods

Lines 167-171. I could not find above this line what “CESD(R)” might mean. Are these 2-sentences about the mean score and reliability in the validation study (reference 32) or in the current survey? If the latter, the information does not belong here but in Results. If the former, set the information off by isolating it in a paragraph of its own and refer to reference 32 again at its end.

Lines 172-179. Same issues as for lines 167-171.

Line 180   Not “included a number of 11 items” but “included 11 items”?

Lines 181-192. For each item on this list, parenthesize all of the subcategories you used or delete the few of them that are now listed, e.g., “pregnancy type (Unplanned, Planned)”

194 – Phrase at the end of sentence: you must, if it’s true, state in a full sentence that informed consent was obtained in advance with a form approved by an ethics committee.

Results

Tables: for this English-language journal, change all decimals from commas to periods.

Table 1. Do you have any more information about the “unmarried/divorced” subgroup, especially living with partner/not living with a partner? The % not married seems extremely high.

Table 2   Please add subcategories to the rows on frequency of “Clinical” depression/anxiety on the Hamilton scales. 8–17 mild depression, 18– 25 moderate depression, >26 severe depression.  Add such severity categories for the HAMA: “less than 17 indicates mild severity, 18–24 mild-moderate severity, and ≥25 moderate severe.[12]”  This is a quote from https://www.ncbi.nlm.nih.gov/pmc/articles/PMC8719567/    where “12” = https://pubmed.ncbi.nlm.nih.gov/2963053/   Otherwise, the 88.37% value for Anxiety and the 84.65% value for Depression seem incredibly high.

Around lines 230-240, add a sentence or two on the % of patients with moderate or worse anxiety per HAMA and moderate or worse depression per HAMD.

Discussion

needs an addition on the limits of the survey: this is not a random population sample but a subgroup selected by admission to hospital. Why were these women hospitalized, and what could their obstetric or other medical problems that caused hospitalization contribute to biasing the survey?

Author Response

Dear Reviewer,

Thank you very much for your observations.  Please find below the answers to your observations.

2.3 Methods

Lines 167-171. I could not find above this line what “CESD(R)” might mean. Are these 2-sentences about the mean score and reliability in the validation study (reference 32) or in the current survey? If the latter, the information does not belong here but in Results. If the former, set the information off by isolating it in a paragraph of its own and refer to reference 32 again at its end.

We made the necessary changes in the manuscript.

Lines 172-179. Same issues as for lines 167-171.

We made the necessary changes in the manuscript.

Line 180   Not “included a number of 11 items” but “included 11 items”?

Thank you very much for your observation. We changed in the manuscript.

Lines 181-192. For each item on this list, parenthesize all of the subcategories you used or delete the few of them that are now listed, e.g., “pregnancy type (Unplanned, Planned)”

Thank you very much for your observation. We changed in the manuscript.

194 – Phrase at the end of sentence: you must, if it’s true, state in a full sentence that informed consent was obtained in advance with a form approved by an ethics committee.

Informed consent was obtained from all participants prior to their inclusion in the study. This ethical approach ensured that participants were fully informed and empowered to make an informed decision about their participation in the study, and that their privacy and confidentiality were protected throughout the research process.

Results

Tables: for this English-language journal, change all decimals from commas to periods.

We made the changes in accordance with your suggestion.

Table 1. Do you have any more information about the “unmarried/divorced” subgroup, especially living with partner/not living with a partner? The % not married seems extremely high.

Civil status was not found to be a risk factor for depression and anxiety, despite 85.12% being divorced/unmarried (Table 1). This can be explained by the evolution of Romanian society and the changing role of women in it: the change in values and social perceptions about family and marriage, the increase in women's financial independence, more young women choosing to have children before getting married compared to previous generations, a desire to avoid potential relationship problems, and some may consider marriage as an outdated social convention and not a priority in their lives.

Table 2   Please add subcategories to the rows on frequency of “Clinical” depression/anxiety on the Hamilton scales. 8–17 mild depression, 18– 25 moderate depression, >26 severe depression.  Add such severity categories for the HAMA: “less than 17 indicates mild severity, 18–24 mild-moderate severity, and ≥25 moderate severe.[12]”  This is a quote from https://www.ncbi.nlm.nih.gov/pmc/articles/PMC8719567/    where “12” = https://pubmed.ncbi.nlm.nih.gov/2963053/   Otherwise, the 88.37% value for Anxiety and the 84.65% value for Depression seem incredibly high.

Thank you very much for your observation. We made the necessary changes in the manuscript.

Around lines 230-240, add a sentence or two on the % of patients with moderate or worse anxiety per HAMA and moderate or worse depression per HAMD.

Thank you very much for your observation. We made the necessary changes in the manuscript.

Discussion

needs an addition on the limits of the survey: this is not a random population sample but a subgroup selected by admission to hospital. Why were these women hospitalized, and what could their obstetric or other medical problems that caused hospitalization contribute to biasing the survey?

We changed most of the Discussion part, in accordance with your observation.

    Thank you very much for your time in reviewing our manuscript!

Round 2

Reviewer 2 Report

Dear authors, 

thank you for your work, however many improvements have to be done

1. The title should be for example : Prevalence and risk factors of depression and anxiety in Pregnant women: A cross-sectional Study

2. The objective should be more specific : I think you should add: in a cohort of pregnant Romanian  women

3. the results presented in the text should have the p-values along with the OR, 95CI, for example"(OR=0.904, 95%CI: 0.826-0.991, p= ?)."

4. Also, these ORs should be mentioned in the abstract and remove the numbers in the abstract introduction  "10 to 29.6%" as these are not necessary in the abstract

5. Also in Table 1, why using uppercase for all the factors? 

6. The conclusion should be shorter and more concise

Author Response

Dear Reviewer,

   Thank you very much for your revision.   We made the corrections as suggested by you. Please be so kind to find the modifications made in the manuscript in red color.

  1. The title should be for example : Prevalence and risk factors of depression and anxiety in Pregnant women: A cross-sectional Study

Thank you very much for your suggestion: Prevalence and risk factors of depression and anxiety among women in the last trimester of pregnancy: a cross-sectional study

  1. The objective should be more specific : I think you should add: in a cohort of pregnant Romanian  women

The main objective of this research was to conduct an antenatal screening in a cohort of pregnant women hospitalized for delivery, in order to assess the prevalence of anxiety and depression. The secondary objective was to identify the risk factors associated with depression and anxiety in women in the third trimester of pregnancy.

  1. the results presented in the text should have the p-values along with the OR, 95CI, for example"(OR=0.904, 95%CI: 0.826-0.991, p= ?)."

Thank you very much for your observation: we made the change in the manuscript.

  1. Also, these ORs should be mentioned in theabstract and remove the numbers in the abstract introduction  "10 to 29.6%" as these are not necessary in the abstract

The results obtained showed that age and the environment of origin are the strongest predictors of mental health in pregnancy period (OR=0.904, 95%CI: 0.826-0.991; p=0,029). Women from Urban area there is an increased probability of falling at a higher level on the dependent variable (moderate depression) (OR=2.454, 95%CI: 1.086-5.545; p=0.032).

  1. Also in Table 1, why using uppercase for all the factors? 

 Thank you very much for the observation, we changed in the Table, in the manuscript.

  1. The conclusion should be shorter and more concise

The study underscores the significance of monitoring mental health throughout pregnancy, as well as identifying pertinent risk factors in order to deliver appropriate care to expectant mothers

The manifestation of both anxiety and depression during pregnancy is often linked with several factors. Recognizing these associated factors can facilitate early implementation of mental health monitoring measures throughout pregnancy.

Demographic and health-related factors are closely interrelated and can have a significant impact on the maternal psycho-affective state. Upon examining the associated risk factors, it was revealed that younger maternal age is associated with prenatal depressive symptoms.

The results obtained showed that age and the environment of origin are the strongest predictors of mental health during pregnancy.

The introduction of a screening program that also provides psychological counseling in prenatal care can be very helpful, for women who are experiencing stress and anxiety or who are in disadvantaged social situations. This program could be implemented at the level of the health system, and medical staff could be trained in providing appropriate counseling and support to these women.

Moreover, the findings emphasize the necessity for implementing interventions aimed at supporting the mental well-being of pregnant women.